# Chemical and Biological Aspects of Different Species of the Genus *Clinanthus* Herb. (Amaryllidaceae) from South America

**DOI:** 10.3390/molecules28145408

**Published:** 2023-07-14

**Authors:** María Lenny Rodríguez-Escobar, Luciana R. Tallini, Julia Lisa-Molina, Strahil Berkov, Francesc Viladomat, Alan Meerow, Jaume Bastida, Laura Torras-Claveria

**Affiliations:** 1Departament de Biologia, Sanitat i Medi Ambient, Facultat de Farmàcia i Ciències de l’Alimentació, Universitat de Barcelona, Av. Joan XXIII 27-31, 08028 Barcelona, Spain; mrodries116@alumnes.ub.edu (M.L.R.-E.); ruscheltallini@ub.edu (L.R.T.); 2.98julialisa@gmail.com (J.L.-M.); fviladomat@ub.edu (F.V.); 2Faculdade de Farmácia, Universidade Federal do Rio Grande do Sul, Av. Ipiranga 2752, Porto Alegre 90610-000, RS, Brazil; 3Department of Plant and Fungal Diversity, Institute of Biodiversity and Ecosystem Research at the Bulgarian Academy of Sciences, 23 Acad. G. Bonchev Str., 1113 Sofia, Bulgaria; berkov_str@yahoo.com; 4School of Life Sciences, Arizona State University, Tempe, AZ 85282, USA; ameerow@asu.edu

**Keywords:** *Clinanthus* sp., *Clinanthus incarnatus*, *Clinanthus variegatus*, Amaryllidaceae alkaloids, alkaloid profiling, Alzheimer’s disease

## Abstract

The genus *Clinanthus* Herb. is found in the Andes Region (South America), mainly in Peru, Ecuador, and Bolivia. These plants belong to the Amaryllidaceae family, specifically the Amaryllidoideae subfamily, which presents an exclusive group of alkaloids known as Amaryllidaceae alkaloids that show important structural diversity and pharmacological properties. It is possible to find some publications in the literature regarding the botanical aspects of *Clinanthus* species, although there is little information available about their chemical and biological activities. The aim of this work was to obtain the alkaloid profile and the anti-cholinesterase activity of four different samples of *Clinanthus* collected in South America: *Clinanthus* sp., *Clinanthus incarnatus*, and *Clinanthus variegatus*. The alkaloid extract of each sample was analyzed by gas chromatography coupled with mass spectrometry (GC-MS), and their potential against the enzymes acetyl- and butyrylcholinesterase were evaluated. Thirteen alkaloids have been identified among these species, while six unidentified structures have also been detected in these plants. The alkaloid extract of the *C. variegatus* samples showed the highest structural diversity as well as the best activity against AChE, which was likely due to the presence of the alkaloid sanguinine. The results suggest this genus as a possible interesting new source of Amaryllidaceae alkaloids, which could contribute to the development of new medicines.

## 1. Introduction

*Clinanthus* Herb., a plant genus of the subfamily Amaryllidoideae (Amaryllidaceae), is endemic to the Andes Region (South America). This plant genus is present in southern Ecuador through Peru, Bolivia, and northern Chile, occupying coastal loma and mountain habitats in both moist and seasonally dry regions [1]. Peru is considered the diversity center of this genus, with 24 species currently recognized [2]. The majority of these species are found in north and central Andean Peru, mainly growing in seasonally dry shrubland or grassy vegetation. As a rule, they are found above 2000 m elevation, and many of them are known only from single localities.

The genus *Clinanthus* was segregated from *Stenomesson* Herb. by Meerow and co-workers, who demonstrated be means of nrDNA ITS sequences that the latter was polyphyletic [3]. This genus represents the lorate or linear-leafed former members of the genus *Stenomesson*, all of which have pseudo-petiolate leaves and are now placed in the tribe Eucharideae [3,4].

Several erstwhile species of the genus *Clinanthus* were recently transferred into *Paramongaia* Velarde [5], the sister genus to *Clinanthus*. This genus is classified as belonging to the tribe Clinantheae, along with *Paramongaia* and *Pamianthe* Stapf [5,6], and it is separated into two sister subclades, discussed by Esquerre-Ibañez and Meerow [1], that diverged about 13 Mya [6]. The larger-flowered of the two conforms to *Stenomesson* subg. *Fulgituba* Ravenna, as conceived in his polyphyletic concept of that genus [7]. The species of this subclade, which usually occur in a diversity of habitats, typically have lorate leaves, generally > 1 cm wide, long funnelform–tubular flowers, long tubes relative to the limb, and various patterns of green in the relatively short limb [1]. The second subclade typically has narrowly lorate to linear leaves < 1 cm wide, usually with no green on the tepals. They appear younger than the first group [6] and include the alpine species *Clinanthus humilis*, which occurs at over 4000 m elevation and maintains its scape inside the bulb until seed capsule maturity [8]. This genus may ultimately number over two dozen or more species [1,2,9,10].

There has been widespread historical confusion between *Clinanthus incarnatus* (Kunth) Meerow and *Clinanthus variegatus* (Ruiz and Pavon) Meerow, with the latter name often applied to herbarium collections of the former (unpubl. herbarium data). The staminal cup length always separates the two species, as does the perigone color and distributional range. *C. variegatus* consistently has a longer staminal cup (ca. 1 cm) than *C. incarnatus*, and the perigone color varies from pale to deep yellow, sometimes suffused with orange-pink (Figure 1). Its flowers can be distinguished by longer perigones than *C. incarnatus* with less dilated tubes, as well as its more southerly distribution.

*Clinanthus incarnatus* (Figure 1A,B) is quite variable and widespread in northern Peru, also occurring rarely in southern Ecuador [8,11]. Herbarium collections suggest that occurrences of this species are concentrated in northern Peru above 2000 m (Andean and inter-Andean valleys), and may extend to central Peru. Throughout its range, *C. incarnatus* varies in flower number, habit, color, and size. Ecuadorean collections tend to be a rich, brick-red color, while Peruvian material ranges from brick-red and red-orange to salmon and pink, but never yellow.

Some authors have described the basionym of *C. variegatus*, *Pancratium variegatum* Ruiz and Pavón (1802), from material they encountered in cultivation in Lima, and were unsure of its actual provenance [12]. The species, as is currently understood, does not appear to occur north of the Department of Cuzco, and ranges from southern Peru to northern Bolivia. Yellow is the primary floral color in their description (“luteo, roseo, albo-viridique colore variegati”). No type specimen exists for this species, as it was presumably lost at sea along with other collections by Ruiz and Pavon. A color plate (Figure 1E) was produced for Ruiz and Pavon (1802), but was never published (the original is held in the archives of the Real Jardín Botánico—CSIC in Madrid, Spain).

The subfamily Amaryllidoideae possesses economic importance, since many species are cultivated as ornamentals due to the great beauty of their flowers. In addition, over the years, they have been and continue to be used for medical purposes owing to the therapeutic properties given by their alkaloids, which are able to biosynthesize. These alkaloids, known as Amaryllidaceae alkaloids, are exclusively synthesized by this subfamily. They have demonstrated extensive biological activities, such as antiviral, antifungal, antibacterial, antimalarial, insecticide, cytotoxic, antitumor, antifungal, antiplatelet, hypotensive, emetic, and acetylcholinesterase (AChE)-inhibitory activities, among others [13,14,15,16,17,18,19,20].

Although chemical and pharmacological interest in the genus *Clinanthus* has increased in recent years, only seven published articles about this genus can be found in the literature, and five of them correspond to the description of new species [1,9,10,21,22]. The first report of a chemical characterization and bioactivity evaluation of a species from this genus was carried out by Adessi and co-workers [23], who reported the inhibition of AChE and BChE enzymes by the alkaloid 6-hydroxymaritidine, extracted and purified from *C. microstephium* [23]. Another study from 2021 reported the chemical characterization and the antiplasmodial activity of two *Clinanthus* species (*C. incarnatus* and *C. ruber*) [24].

To the best of our knowledge, the present study represents the first report evaluating the inhibition of AChE and BuChE of different species from the genus *Clinanthus*. Furthermore, it presents the chemical characterization of three different species which may contribute to properly identifying species from this genus as well as the differentiation between *C. incarnatus* and *C. variegatus*, two species which have previously been confused. 

The aim of the present study was to extract and characterize the alkaloids present in three species of the *Clinanthus* genus (*C. incarnatus*, *C. variegatus*, and *Clinanthus* sp.), and to detect the main skeleton types and characteristic alkaloids of this genus. Furthermore, the alkaloid extracts of the three species will be evaluated in terms of bioactivity and inhibition of AChE and BuChE enzymes in order to identify possible species that could represent new sources of cholinesterase inhibitors for the palliative treatment of Alzheimer’s disease.

## 2. Results and Discussion

### 2.1. Alkaloid Identification and Quantification

Thirteen alkaloids have been identified in the *Clinanthus* species by GC-MS (Table 1), the structures of which are represented in Figure 2. Six compounds have not been identified among these *Clinanthus* extracts, which seems to be an interesting source of new Amaryllidaceae alkaloids. Every structure detected in the alkaloid extract of these samples has been quantified and reported as μg Gal·100 mg^−1^ DW.

GC/MS is the most often used methodology for identifying Amaryllidaceae alkaloids due to its ability to quickly recognize specific compounds in a plant extract or to isolate pure compounds without a derivatization step. It is also highly sensitive and allows for a correct volatilization of this type of alkaloids under gas chromatographic conditions. The ionization method usually used is electron impact (EI), which provides extensive fragmentation of the molecule by undergoing a highly energetic electron beam, usually with an energy of 70 eV [25]. 

The Amaryllidaceae alkaloids share a biogenetic relationship, and although more complex classification systems can be found, they can be classified into nine fundamental skeleton types: norbelladine-, lycorine-, homolycorine-, crinine-, haemanthamine-, narciclasine-, tazettine-, montanine-, and galanthamine-type alkaloids [17,26]. These alkaloid scaffolds often show a GC-MS fragmentation pattern that has been specially described for galanthamine-, lycorine-, and homolycorine-type structures [27,28,29]. Usually, nuclear magnetic resonance (NMR) and circular dichroism (CD) analyses are required in order to confirm the stereochemistry of the alkaloids belonging to the haemanthamine/crinine-type. Some structures belonging to this skeleton type have been described herein as isomers. 

All the samples evaluated in this work demonstrated the presence of lycorine-type structures among their alkaloid profiles, especially *Clinanthus* sp., sample A, which presented a total of 71.9 μg Gal·100 mg^−1^ DW of compounds with this scaffold. Different studies have shown that lycorine-type alkaloids have antiviral activity against the influenza A (H5N1) virus [17,30,31], dengue virus, polio virus, and hepatitis [32,33]. Furthermore, some of these alkaloids have antitumor activity against highly invasive ovarian cancer, colorectal adenocarcinoma [34,35], prostate cancer, pancreatic cancer, and lung cancer [36,37,38], in addition to astrocytoma, glioma, human myetitic leukemia, and hepatocarcinoma [39,40]. Sample A is also characterized by the absence of alkaloids from haemanthamine-, homolycorine-, and galanthamine-types.

Both extracts of *C. variegatus*, samples C and D, showed a high concentration of haemanthamine/crinine-type alkaloids, totalizing 67.7 and 110.8 μg Gal·100 mg^−1^ DW, respectively. They were mainly represented by the alkaloids 8-*O*-demethylmaritidine (6) and haemanthamine (8) in both samples. Among the plants studied in this work, *C. variegatus* was the only species that presented this alkaloid-type skeleton in its content. Haemanthamine/crinine-type alkaloids have presented good activity against the proliferation of ovarian, lung, breast, colon, and brain cancer cell lines [39,40,41,42,43]. Moreover, they have also exhibited antimalarial activity [44,45] and antiviral activity against the influenza A (H5N1) virus [31]. 

Although in low concentrations, trisphaeridine, a narciclassine-type alkaloid, was observed in all the samples evaluated in this work except *C. incarnatus* (see Table 1). The galanthamine- and homolycorine-type scaffolds were been identified only in the extracts of the species *C. variegatus* (samples C and D). Moreover, these two samples of *C. variegatus* presented very similar alkaloid profiles between them, as well as the highest quantity of total alkaloids, with 141.9 and 224.0 μg Gal·100 mg^−1^ DW, respectively (see Table 1). Sample D showed the highest content of galanthamine-type compounds, with a total of 26.5 μg Gal·100 mg^−1^ DW, sanguinine being the most representative alkaloid of this group with 19.4 μg Gal·100 mg^−1^ DW. No scientific report regarding the chemical profiling of the species *C. variegatus* has been found in the literature.

Galanthamine, a galanthamine-type alkaloid, is the most relevant Amaryllidaceae alkaloid, and it has been used for the palliative treatment of Alzheimer’s disease since its approval by the FDA in 2001. It acts as a powerful selective, competitive, and reversible inhibitor of the enzyme acetylcholinesterase (AChE). Furthermore, it is able to act at the level of nicotinic receptors, facilitating the release of acetylcholine (ACh) and, thus, managing to attenuate the symptomatology of neurodegenerative Alzheimer’s [46]. Some homolycorine-type alkaloids show antimalarial activity against *Plasmodium falciparum* and against *Trypanosoma cruzi* [47,48,49]. *C. variegatus* samples present a homolycorine-type alkaloid, hippeastrine. This alkaloid has recently demonstrated selective activity against the parasite *Trypanosoma cruzi*, which is responsible for the Chagas disease [50].

*Clinanthus* sp. and *C. incarnatus* were the samples with the lowest alkaloid contents, and no alkaloids from haemanthamine/crinine-, galanthamine-, or homolycorine-types were found in these species. Besides the high concentration of lycorine-type alkaloids quantified in *Clinanthus* sp., low amounts of trisphaeridine, a narciclassine-type alkaloid, and an unidentified structure, UI 18, were also detected in this sample. The alkaloid composition of *C. incarnatus* was dominated by the presence of unidentified structures, totaling 90.1 μg Gal·100 mg^−1^ DW, which suggests this species as a potential source of new Amaryllidaceae alkaloids. The structures UI 18 and UI 19 were the most representative among the unidentified compounds (see Table 1), and, according to their fragmentation pattern, both molecules probably belong to lycorine-type scaffolds. Soto-Vásquez and co-workers evaluated the alkaloid content of *C. incarnatus* collected in Peru [24]. The authors described the presence of lycorine-, haemanthamine/crinine-, and galanthamine-type alkaloids in this species, the first group being the most representative [24]. 

### 2.2. Cholinesterase Inhibitory Activity

All of the *Clinanthus* extracts had their in vitro inhibitory potential against acetyl- and butyrylcholinesterase evaluated herein. According to the results represented in Figure 3, the best cholinesterase activity was obtained with the extracts of *C. variegatus*, samples C and D, which presented high potential against AChE with IC_50_ values of 1.23 ± 0.17 and 0.80 ± 0.02 μg·mL^−1^, respectively. The alkaloid galanthamine, used as the control, showed IC_50_ values of 0.22 ± 0.01 μg·mL^−1^ against AChE.

The presence of the alkaloid sanguinine in samples C and D must be correlated with the high IC_50_ values found in these extracts. In vitro experiments have shown that this alkaloid is more active than galanthamine against AChE; however, it does not present the same ability as galanthamine to cross the blood–brain barrier [51]. In silico molecular docking experiments suggest that sanguinine possess two hydrogen bonds with His447 and Ser203 towards AChE (4EY7 structure), while galanthamine holds just one hydrogen bond with Ser203, which could explain the superior in vitro AChE-inhibitory activity of sanguinine over galanthamine against this enzyme [45]. Samples A and B showed higher values of IC_50_ against AChE (7.79 ± 0.57 μg·mL^−1^ and 57.76 ± 2.88 μg·mL^−1^, respectively), thus indicating that the corresponding species are not so active in inhibiting this enzyme. These data correlate with the lower alkaloid content of these two samples. 

## 3. Materials and Methods

### 3.1. Plant Material

Three species of the genus *Clinanthus* Herb., totaling four samples (A, B, C, and D) collected from different geographic areas of Ecuador, Perú, and Bolivia, were studied herein. All of the samples were identified by Prof. Alan Meerow (Arizona State University, Tempe, AZ, USA). Sample A, *Clinanthus* sp., was collected in Junín (Perú) by Dr. Geoffrey Herklots; specimen voucher: 26 April 2017, Meerow 3516 (NA). Sample B, *Clinanthus incarnatus* (Kunth) Meerow, was collected in Ecuador by Ray Baker; specimen voucher: 15 May 2015, Meerow 3518 (NA). Sample C, *Clinanthus variegatus* (Ruiz and Pavon) Meerow, was collected in La Paz (Bolivia) by Fred Meyer; specimen voucher: 20 March 1999, Meerow 1159 (FTG). Finally, sample D, *Clinanthus variegatus* (Ruiz and Pavon) Meerow was collected in Sandia (Peru); specimen voucher: 12 April 2009, Meerow 3521 (NA). 

### 3.2. Alkaloid Extraction

The plant material used in the present study consisted of bulbs. They were cut and dried at 40 °C, then milled with a rotary blade mill (stainless steel grinder, Taurus) into a fine powder. The powder was macerated with methanol at room temperature for 3 days, changing the solvent daily (3 × 50 mL) and applying 20 min of an ultrasonic bath 4 times per day. The mash was filtered and evaporated to dryness under reduced pressure. The crude extracts were acidified with an aqueous solution of sulfuric acid (2%, *v*/*v*) to pH 2, then cleaned with diethyl ether to remove neutral material. The aqueous solution was basified with ammonium hydroxide at 25% (*v*/*v*) to pH 9–10, then finally extracted with ethyl acetate to obtain the alkaloid extract (AE).

### 3.3. GC-MS Analysis

Two mg of each sample were dissolved in 1 mL of methanol containing 0.025 μg·mL^−1^ of codeine, used as the internal standard, and injected into a gas chromatograph (Agilent Technologies 6890N, Agilent Technologies, Santa Clara, CA, USA) coupled with mass spectrometry (Agilent Technologies 5975, Agilent Technologies, Santa Clara, CA, USA), both of which were obtained from Hewlett Packard, Palo Alto, CA, USA. The equipment was operated using electronic impact ionization (EI) at 70 eV with an autoinjector, 7683B Series (Agilent Technologies, Santa Clara, CA, USA). The column was a Tecknokroma TR-45232 Sapiens-X5MS (30 m × 0.25 mm, film thickness 0.25 µm), and the splitless injection volume was 1 µL. The temperature gradient was as follows: 12 min at 100 °C, 100–180 °C at 15 °C/min, 1 min of retention at 180 °C, 180–300 °C at 5 °C/min, and 10 min of holding at 300 °C. The injector and detector temperatures were 250 and 280 °C, respectively. The carrier gas (He) flow rate was 1 mL/min.

### 3.4. Alkaloid Identification

The alkaloids were identified by comparison of their mass spectra and their Kovats Retention Indices (RI) with those of authentic standards from the Natural Products Research Group Amaryllidaceae alkaloids library of the Faculty of Pharmacy (University of Barcelona), which contains more than 300 Amaryllidaceae alkaloids. This database, which is periodically updated, was built with Amaryllidaceae alkaloids isolated from Amaryllidaceae plants by our research group during its 40 years of research. Their structures were elucidated using various spectroscopic techniques, such as NMR, UV, CD, and MS. The mass spectra were deconvoluted using AMDIS 2.64 (Automatic Mass spectral Deconvolution and Identification System (NIST)) software. The RI values of the compounds were measured in relation to a standard n-hydrocarbon calibration mixture (C9–C36) [52].

### 3.5. Alkaloid Quantification 

A relative quantification of the alkaloids was performed. The Amaryllidaceae alkaloids were quantified with a galanthamine calibration curve (at 10, 20, 40, 60, and 80 μg·mL^−1^) using codeine (0.025 μg·mL^−1^) as the internal standard. The area of deconvoluted peaks was taken for quantification. Although it is not an absolute quantification, it is considered suitable for comparing the quantity of specific alkaloids between samples and with other analyses and Amaryllidaceae plants already quantified using this method [52]. The results are expressed as μg of galanthamine (Gal) per 100 mg of dry weight (DW) of the plant (μg Gal·100 mg^−1^ DW). The calculation analysis was performed using Excel 2016 software.

### 3.6. AChE and BuChE Inhibition Assay

The inhibitory activities of the cholinesterases were determined according to Ellman and co-workers [53], with some modifications, as by López and co-workers [54]. Stock solutions with 518U of AChE from *Electrophorus electricus* (Merck, Darmstadt, Germany) and BuChE from equine serum (Merck, Darmstadt, Germany), respectively, were prepared and kept at −20 °C. Acetylthiocholine iodide (ATCI), *S*-butyrylthiocholine iodide (BTCI), and 5,5-dithiobicis (2-nitrobenzoic acid) (DTNB) were obtained from Merck (Darmstadt, Germany). Fifty microliters of AChE or BuChE (both enzymes were used at 6.24 U) in phosphate buffer (8 mM K_2_HPO_4_, 2.3 mM NaH_2_PO_4_, 0.15 NaCl, pH 7.5) and 50 µL of the sample dissolved in the same buffer were added to the wells. The plates were incubated for 30 min at room temperature. Then, 100 µL of the substrate solution (0.1 M Na_2_HPO_4_, 0.5 M DTNB, and 0.6 mM ATCI or 0.24 mM BTCI in Millipore water, pH 7.5) was added. These reagents were obtained from Merck (Darmstadt, Germany). After 10 min, the absorbance was read at 405 nm on a Labsystem microplate reader (Helsinki, Finland). The enzymes’ activities were calculated as percentages compared to a control, which used a buffer without any inhibitor. Galanthamine served as a positive control (using the following concentrations for AChE: 0.1, 0.25, 0.3, 0.5, and 2.5 µg·mL^−1^; and for BuChE: 1, 4, 8, 10, and 15 µg·mL^−1^). The calibration curves of samples A, B, C, and D (1, 5, 10, 15, 20, 30, and 40 µg·mL^−1^; 10, 15, 20, 30, 40, 60, and 100 µg·mL^−1^; 0.1, 0.5, 1, 2, 3, 5, and 10 µg·mL^−1^; and 0.1, 0.5, 1, 2, 3, 5, and 10 µg·mL^−1^, respectively) were applied to obtain the IC_50_ values against the AChE enzyme, and the following curves were used to obtain the IC_50_ values against BuChE: 50, 75, 100, 125, 150, 200, and 250 µg·mL^−1^; 60, 100, 200, 250, 350, 500, and 1000 µg·mL^−1^; 25, 50, 100, 150, 200, 250, and 300 µg·mL^−1^; and 25, 50, 100, 150, 200, 250, and 300 µg·mL^−1^, respectively. The inhibitory data regarding the cholinesterases were analyzed using Prism 9 software.

## 4. Conclusions

The alkaloid profiles and the cholinesterase inhibitory potential of four different samples of *Clinanthus* collected in South America were obtained herein, with thirteen identified alkaloids and six unidentified ones from haemanthamine- and lycorine-types. The chemical characterization of *C. incarnatus* and *C. variegatus* presented herein may contribute to these species’ differentiation. This is the first chemical and biological report about the species *C. variegatus*, which showed interesting structural diversity and high activity against AChE, a relevant enzyme in the cholinergic therapy for Alzheimer’s disease. The GC-MS results, especially those of the species *C. incarnatus*, suggest that this genus could be a possible new source of Amaryllidaceae alkaloids, which could be an interesting finding due to the pharmaceutical importance of this plant subfamily.

## Figures and Tables

**Figure 1 molecules-28-05408-f001:**
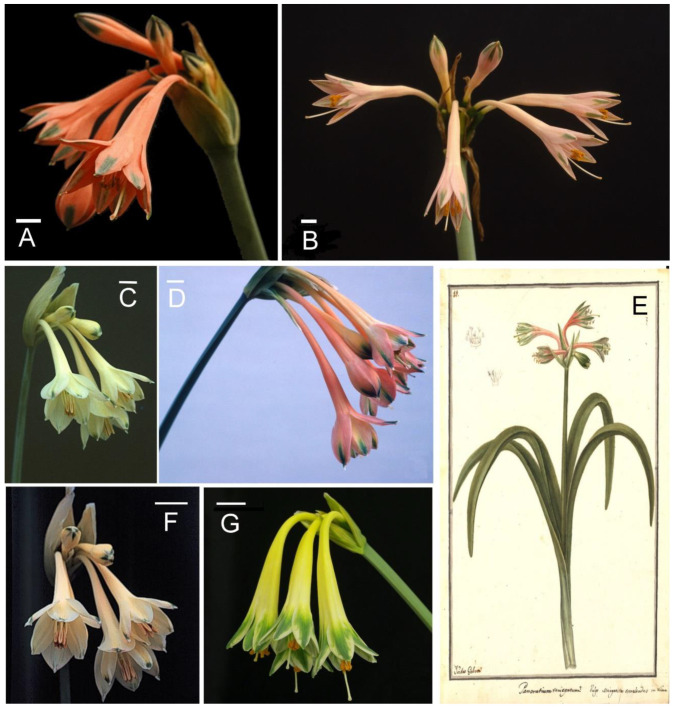
*Clinanthus* species analyzed in this publication. (**A**,**B**) *Clinanthus incarnatus* from Ecuador; (**C**,**E**,**F**) *Clinanthus variegatus* from Peru; (**D**) *Clinanthus variegatus* from Bolivia; (**G**) *Clinanthus* sp. from Peru. All scale bars = 1 cm. The source of the photos was Prof. Alan Meerow.

**Figure 2 molecules-28-05408-f002:**
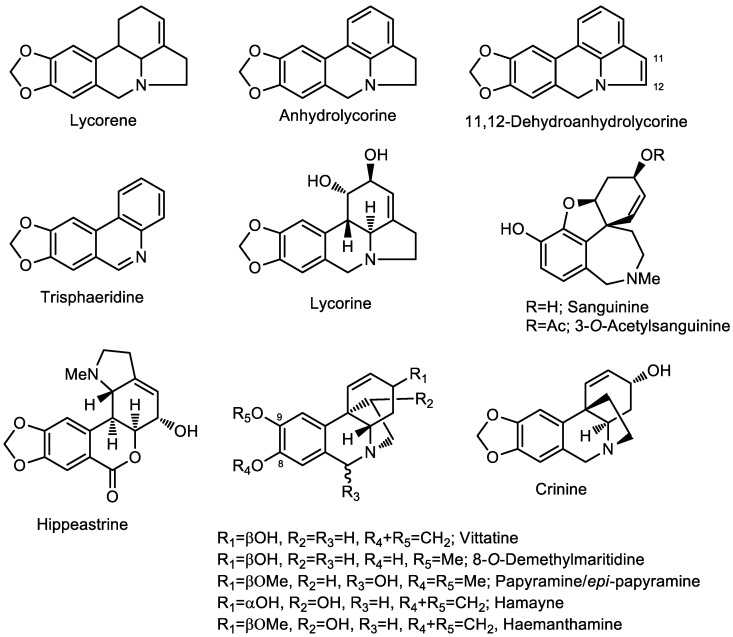
Amaryllidaceae alkaloids identified in *Clinanthus* Herb. extracts by GC-MS.

**Figure 3 molecules-28-05408-f003:**
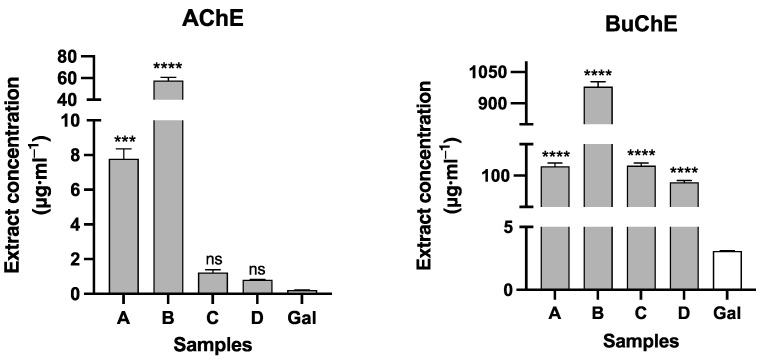
AChE- and BuChE-inhibitory activity of *Clinanthus* Herb. extracts. A: *Clinanthus* sp.; B: *Clinanthus incarnatus*; C: *Clinanthus variegatus*; D: *Clinanthus variegatus*; Gal: galanthamine. Data are expressed as the means ± standard deviation (SD) of three independent experiments. **** and *** represent significant difference versus the control (Gal) with AChE. *p* < 0.0001, *p* < 0.001, respectively, and ns (no significant). One-way ANOVA was conducted with Dunnet’s multiple comparison test (the differences are with respect to the result of galanthamine with AChE and BuChE).

**Table 1 molecules-28-05408-t001:** Alkaloid profiling of different species of *Clinanthus* Herb. by GC-MS. Values are expressed in μg Gal·100 mg^−1^ DW.

Alkaloid	RI	MS	A	B	C	D
**Lycorine-type (total)**			**71.9**	**39.5**	**29.5**	**40.3**
Lycorene (1)	2245.2	255 (60), 254(100), 227 (17), 226 (19), 211 (13), 183 (11)	-	-	16.4	12.2
Anhydrolycorine (2)	2519.6	251 (46), 250 (100), 192 (12), 191 (11), 124 (7), 95 (7)	39.6	-	-	7.7
11,12-Dehydroanhydrolycorine (3)	2613.9	249 (61), 248 (100), 191 (10), 190 (25), 189 (7), 95 (15)	26.1	20.7	6.0	7.1
Lycorine (4)	2735.2	287 (27), 286 (18), 268 (22), 250 (19), 227 (67), 226 (100)	6.2	18.8	7.1	13.3
**Haemanthamine/crinine-type (total)**			**-**	**-**	**67.7**	**110.8**
Vittatine/crinine (5a/5b)	2467.2	271 (100), 200 (24), 199 (72), 187 (64), 128 (25), 115 (26)	-	-	7.0	10.5
8-*O*-Demethylmaritidine (6)	2501.3	273 (100), 230 (28), 202 (25), 201 (82), 189 (55), 175 (23)	-	-	24.8	44.8
Papyramine/*epi*-papyramine (7a/7b)	2545.5	317 (100), 286 (50), 259 (46), 230 (83), 187 (36), 186 (43)	-	-	4.7	6.6
Haemanthamine (8)	2628.9	301 (17), 273 (23), 272 (100), 242 (18), 240 (20), 181 (28)	-	-	26.3	35.8
Hamayne (9)	2698.8	287 (5), 259 (20), 258 (100), 214 (8), 211 (16), 186 (11)	-	-	4.9	13.1
**Galanthamine-type (total)**			**-**	**-**	**14.2**	**26.5**
Sanguinine (10)	2429.9	273 (100), 272 (81), 256 (20), 202 (37), 160 (44), 115 (19)	-	-	9.9	19.4
3-*O*-Acetylsanguinine (11)	2515.0	315 (52), 256 (100), 255 (60), 254 (40), 212 (27), 96 (60)	-	-	4.3	7.1
**Narciclassine-type (total)**			**3.6**	**-**	**4.6**	**5.2**
Trisphaeridine (12)	2304.1	223 (100), 222 (36), 164 (15), 138 (22), 137 (10), 111 (13)	3.6	-	4.6	5.2
**Homolycorine-type (total)**			**-**	**-**	**19.9**	**29.1**
Hippeastrine (13)	2881.2	315 (<1), 162 (5), 126 (14), 125 (100), 124 (15), 96 (51)	-	-	19.9	29.1
**Unidentified Alkaloid (total)**			**8.9**	**90.1**	**6.0**	**12.1**
UI (14) Haemanthamine/crinine-type *	2599.5	315 (100), 286 (9), 272 (27), 256 (89), 254 (57), 218 (58)	-	-	-	6.8
UI (15) Haemanthamine/crinine-type *	2670.1	297 (98), 296 (36), 278 (53), 252 (36), 251 (22), 132 (100)	-	-	6.0	5.3
UI (16) Lycorine-type *	2690.8	373 (60), 342 (59), 250 (31), 225 (53), 212 (100) 131 (51)	-	11.2	-	-
UI (17) Ismine-type *	2732.4	299 (32), 250 (20), 225 (65), 224 (100), 212 (8), 166 (9)	-	14.8	-	-
UI (18) Lycorine-type *	2851.8	279 (79), 278 (100), 280 (14), 263 (9), 235 (16), 178 (12)	8.9	30.5	-	-
UI (19) Lycorine-type *	2854.9	359 (2), 358 (4), 299 (43), 268 (100), 250 (51), 212 (26)	-	33.6		-
**Total Alkaloids**			**84.4**	**129.6**	**141.9**	**224.0**

**A**: *Clinanthus* sp.; **B**: *Clinanthus incarnatus*; **C**: *Clinanthus variegatus* from Bolivia; **D**: *Clinanthus variegatus* from Peru; **MS**: mass spectrum; **RI**: retention index; * proposed structure type according to the fragmentation pattern.

## Data Availability

Data is contained within the article or Appendix A.

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
