# Peer review of "Chemical and Biological Aspects of Different Species of the Genus *Clinanthus* Herb. (Amaryllidaceae) from South America"

_molecules, 2023, doi:10.3390/molecules28145408_

Round 1
Reviewer 1 Report
This study performed the alkaloid profile and anti-cholinesterase activity of four different samples of Clinanthus collected in South America: Clinanthus sp., Clinanthus incarnates, and Clinanthus variegatus. There are some question need resolve.
(1) The chemical constituents of different species of the genus Clinanthus was identified rarely.
(2) The chromatogram of different species of Clinanthus need be added.
(3) The content of pharmacological activity of different species was too simple.
(4) The information of abscissa and ordinate of Figure 3 was missing.
The quality of English language need impove.
Author Response
1.- The identification of the compounds is performed comparing their RI and fragmentation pattern with those of the Amaryllidaceae alkaloids library, built with Amaryllidaceae alkaloids previously isolated and the structures elucidated with spectroscopic techniques as NMR, UV, CD and MS by our group, along its 40 years of research.
We have added this paragraph in Alkaloid Identification Section in the text:
“This database has been built with the Amaryllidaceae alkaloids isolated in Amaryllidaceae plants by our group along its 40 years of research, and the structures elucidated using various spectroscopic techniques such as NMR, UV, CD and MS [51].”
2.- We have added the GC-MS chromatograms of the different samples (A, B, C and D) in a supplementary material.
3.- We have added two new paragraphs in the Results and discussion section explaining in more detail the pharmacological activity.
4.- We have added “Extract concentration” next to the µg·mL -1 in the Figure 3.
Reviewer 2 Report
This article is devoted to the study of various species of the genus Clinanthus Herb. (Amaryllidaceae) from South America. The article is well written, the volume and quality of the data corresponds to the Journal. The relevance is due to the expansion of the use of plant biomass for the production of biologically active substances. Abstract is well written. In the introduction, the main literature data on Clinanthus Herb are presented quite accurately and completely. (Amaryllidaceae). Experimental questions:
1. The chemical composition of plant biomass depends not only on the species and genus of the plant, but also on the place where it grew, climate, soil, distance from roads and other factors. To what extent is this knowledge applicable to what the authors have obtained in this article?
2. Table 1. This table shows the major alkaloid substances included in the composition. Please clarify in the text of the article which minor components are also present.
3. Figure 2. Many different alkaloids are listed. This is good. Where can these alkaloids or their mixture be practically applied? In addition, they have varying degrees of biological activity, this can also be indicated.
4. Why did the authors choose this particular alkaloid extraction method?
5. Conclusions needs to be significantly expanded.
6. Please cite: 10.3390/molecules27186129.
Author Response
1.- Certainly, the chemical composition of plants depends on the species and the genus, but also on the place they grow, the climate, soil, and there are also ontogenic variations of alkaloid composition. In our results, these facts can be reflected in Clinanthus variegatus samples (C and D). Their composition is very similar, but there are slight differences, basically the quantity of alkaloids, or the presence of anhydrolycorine, which only appears in sample D. These slight differences are possibly due to the factors that the reviewer mentions.
2.- The extract analysed by GC-MS and whose alkaloid composition is resumed in table 1 corresponds to a purified alkaloid extract. In this extract fatty acids and other organic impurities have been minimised, as well as other polar impurities that can be present in methanolic extracts such as organic acids, aminoacids, sugar alcohols, mono-, di- and trisaccharides, fatty acids, sterols, phosphates, and other unknown compounds. The development and validation of the method can be found in:
Berkov, S.; Bastida, J.; Viladomat, F.; Codina, C. Development and validation of GC-MS method for rapid determination of galanthamine in Leucojum aestivum and Narcissus ssp.: A metabolomics approach. Talanta 2011, 83: 1455-1465.
3.- We have added two new paragraphs in the Results and discussion section explaining in more detail the pharmacological activity.
4.- The procedure used for the extraction and purification of alkaloids has been performed for several years with slight modifications for a more sustainable and less toxic solvents method. For instance, we have changed the solvent used for extraction to a less toxic one and with fewer extracted subproducts that interfere with the alkaloid ionization and detection in the GC-MS analysis.
Tallini, L.R.; Torras-Claveria, L.; Borges, W.S.; Kaiser, M.; Viladomat, F.; Zuanazzi, J.A.S.; Bastida, J. N-oxide alkaloids from Crinum amabile (Amaryllidaceae). Molecules 2018, 23, 1277.
5.- Conclusions have been extended.
6.- The article recommended to cite is about essential oils research. Maybe there is a possible mistake in the doi number.
Reviewer 3 Report
This manuscript is about the Chemical and biological aspects of different species of the genus Clinanthus Herb. (Amaryllidaceae) from South America, this work is interesting. But some important data of the manuscript is losing. Here some improvements that must be considered:
1. Please improved the GC-MS graph for these herbs
2. As for figure 4, the information in this figure is losing, please improve it.
3. The Alkaloid Quantification part is not complete, please provide the important data
4. The manuscript has mentioned the in silico molecular docking results, the photo of molecular docking should be provided.
5. The whole manuscript should be improved.
Author Response
1.- We have added the GC-MS chromatograms of the different samples (A, B, C and D) in a supplementary material.
2.- In the original manuscript there is not a Figure 4. In case the reviewer refers to the original Figure 3, we have improved it as suggested by Reviewer 1.
3.- The quantification method performed in this research is not an absolute quantification. The amount of different alkaloids is expressed as µg GAL/DW. This is a semi-quantitative method, and allows us to compare samples and to detect species with higher amounts of alkaloids.
Torras-Claveria, L.; Berkov, S.; Codina, C.; Viladomat, F.; Bastida, J. Metabolomic analysis of bioactive Amaryllidaceae alkaloids of ornamental varieties of Narcissus by GC-MS combined with k-means cluster analysis. Ind. Crop. Prod. 2014, 56, 211-222.
Lisa-Molina, J.; Gómez-Murillo, P.; Arellano-Martín, I.; Jiménez, C.; Rodríguez-Escobar, M.L.; Tallini, L.R.; Viladomat, F.; Torras-Claveria, L.; Bastida, J. Alkaloid profile in wild autumn-flowering daffodils and their acetylcholinesterase inhibitory activity. Molecules 2023 28 (3):1239.
4.- In silico molecular docking experiments of sanguinine and galanthamine are only mentioned in the text in reference to previous studies performed by Rojas-Vera et al., 2021 (reference number 45).
5.- The whole manuscript has been improved following reviewers 1, 2 and 3 recommendations.
Round 2
Reviewer 3 Report
The paper has improved a lot, which can be accepted in current form